# Prmt5 promotes vascular morphogenesis independently of its methyltransferase activity

Aurélie Quillien[1,2]*, Guerric Gilbert[3], Manon Boulet[1,3], Séverine Ethuin[1], Lucas Waltzer[3], Laurence Vandel[1,3]*

1 Centre de Biologie du Développement (CBD), Centre de Biologie Intégrative (CBI), Université de Toulouse, CNRS, UPS, Toulouse, France, 2 RESTORE, INSERM UMR1301, CNRS UMR5070, Université Paul Sabatier, Université de Toulouse, Toulouse, France, 3 Université Clermont Auvergne, CNRS, INSERM, iGReD, Clermont-Ferrand, France

* aurelie.quillien@gmail.com (AQ); laurence.vandel@uca.fr (LV)

**Data Availability Statement:** All relevant data are within the manuscript and its Supporting Information files.

## Abstract

During development, the vertebrate vasculature undergoes major growth and remodeling. While the transcriptional cascade underlying blood vessel formation starts to be better characterized, little is known concerning the role and mode of action of epigenetic enzymes during this process. Here, we explored the role of the Protein Arginine Methyl Transferase Prmt5 in blood vessel formation as well as hematopoiesis using zebrafish as a model system. Through the combination of different *prmt5* loss-of-function approaches we highlighted a key role of Prmt5 in both processes. Notably, we showed that Prmt5 promotes vascular morphogenesis through the transcriptional control of ETS transcription factors and adhesion proteins in endothelial cells. Interestingly, using a catalytic dead mutant of Prmt5 and a specific drug inhibitor, we found that while Prmt5 methyltransferase activity was required for blood cell formation, it was dispensable for vessel formation. Analyses of chromatin architecture impact on reporter genes expression and chromatin immunoprecipitation experiments led us to propose that Prmt5 regulates transcription by acting as a scaffold protein that facilitates chromatin looping to promote vascular morphogenesis.

## Author summary

Blood vessel formation is an essential developmental process required for the survival of all vertebrates. The vascular anatomy and the mechanisms involved in vessel formation are highly conserved among vertebrates. Hence, we used zebrafish as a model, to decipher the role and the mode of action of Prmt5, an enzyme known to regulate gene expression, in vascular morphogenesis and in blood cell formation *in vivo*. Using different approaches, we highlighted a key role of Prmt5 during both processes. However, we found that while blood cell formation required Prmt5 enzymatic activity, vascular morphogenesis was independent on its activity. Prmt5 has been proposed as a therapeutic target in many diseases, including cancer. Yet, we show here that Prmt5 acts at least in part independently of its methyltransferase activity to regulate vascular morphogenesis. By

**Funding:** This work was supported by the French Muscular Dystrophy Association https://www.afm-telethon.fr (grant 20102) and from the Fondation ARC pour la Recherche Contre le Cancer https://www.fondation-arc.org (grant SFI20121205590) to LV; By the Agence Nationale pour la Recherche (ANR) https://www.anr.fr (grant ANR-17-CE12-0030-03) and the Fondation ARC (grant PJA20171206371) to LW. AQ was the recipient of a post-doctoral fellowship from the Fondation ARC (grant PDF20140601073) and GG got a PhD fellowship from the University Clermont-Auvergne https://www.uca.fr. The funders had no role in study design, data collection and analysis, decision to publish, or preparation of the manuscript.

**Competing interests:** The authors have declared that no competing interests exist.

shedding light on a mechanism of action of Prmt5 that will be insensitive to enzymatic inhibitors, our data calls forth the design of alternative drugs. In addition, this non-canonical function of Prmt5 may have a more pervasive role than previously thought in physiological conditions, *i.e.* during development, but also in pathological situations such as in tumor angiogenesis and certainly deserves more attention in the future.

## Introduction

Blood vessel formation is an essential developmental process required for the survival of all vertebrates and much effort has been devoted to reveal the underlying molecular pathways and to identify key molecules that regulate different aspects of this process. The vascular anatomy and the mechanisms involved in vessel formation are highly conserved among vertebrates (for a review, [1]). Hence, in the past two decades, zebrafish has been proven to be a useful model to study vascular morphogenesis and blood cell formation *in vivo* [2–5].

In vertebrates, blood cell formation is tightly associated with the development of the vascular system. Hematopoietic Stem Cells (HSC), which give rise to the different blood cell lineages, emerge directly from the ventral part of the dorsal aorta, an area referred to as the hemogenic endothelium. Notably, the ETS transcription factor ETV2 functions as a master regulator for the formation of endothelial and hematopoietic cell lineages through the induction of both blood cells and vasculature transcriptional programs, in mouse and in zebrafish [6,7]. In endothelial cells, ETV2 regulates the expression of other ETS transcription factors, VEGF (Vascular Endothelial Growth factor) signaling receptors and effectors, Rho-GTPases and adhesion molecules [6,7]. Besides, adhesion molecules have been shown to be crucial players in vascular morphogenesis as Vascular Endothelial cadherin (VE-cad/ Cdh5) and endothelial cell-selective adhesion molecule (Esama) are essential for junction remodeling and blood vessel elongation in zebrafish [8,9]. Indeed, loss-of-function of both *cdh5* and *esama* leads to the formation of disconnected vessels and delayed lumen formation. Likewise, knock-down of the scaffold protein Amolt2, which associates to VE-cadherin, also leads to sprout elongation defects and narrowed aortic lumen [10]. While the transcriptional cascade underlying blood vessel formation starts to be better characterized, little is known about the role and mode of action of epigenetic enzymes during this process. Even though chromatin-modifying enzymes have been described as central in cardiovascular disease and development [11,12], only few examples illustrate in detail the role of epigenetic enzymes during blood vessel development. For instance, the chromatin-remodeling enzyme BRG1 affects early vascular development as well as hematopoiesis in mice [13], and the histone acetyltransferase P300 has been proposed to be recruited at the promoter of specific endothelial genes by the ETS transcription factor ERG (ETS Related Gene) to control their expression both *in vivo* in zebrafish and in HUVECs (Human Umbilical Vein Endothelial Cells) [14,15].

Given the common origin of blood and endothelial cells, and their partially shared transcriptional programs, it is plausible that known chromatin-modifying enzymes affecting hematopoiesis could also control blood vessel formation. Along this line, the epigenetic enzyme Prmt5 (Protein Arginine Methyltransferase 5) has been identified as a key player in blood cell formation [16], but its impact on endothelial development has not been investigated to date. Prmt5 forms a hetero-octameric complex with MEP50 (Methylosome Protein 50)/ Wdr77 (WD repeat Domain 77) [17], which catalyzes the symmetric di-methylation of arginine residues on a variety of proteins including histones and therefore acts on many cellular processes such as genome organization, transcription, differentiation, cell cycle regulation or

spliceosome assembly [12,18–22]. Prmt5 regulates transcription through the methylation of arginine residues on histones H3 and H4 and has been shown to participate in several differentiation processes such as myogenesis, germ cell differentiation or hematopoiesis [16,23–25]. In mice, *prmt5* knock-out prevents pluripotent cells to form from the inner cell mass and is embryonic lethal [26]. Conditional loss of *prmt5* leads to severe anemia and pancytopenia as Prmt5 maintains Hematopoietic Stem Cells (HSCs) and ensures proper blood cell progenitor expansion by repressing p53 expression [16]. In this context, Prmt5 protects HSCs from DNA damages by allowing the splicing of genes involved in DNA repair [27].

Here, we explored the role of the Protein Arginine MethylTransferase Prmt5 in hematopoiesis and during blood vessel formation in zebrafish. Through loss-of-function approaches by generating a *prmt5* mutant and using *prmt5* morpholinos, we highlighted a key role of Prmt5 in both processes. Interestingly, rescue experiments as well as the use of a drug inhibitor showed that, in contrast with blood cell formation, vascular morphogenesis was independent on Prmt5 methyltransferase activity. Finally, analyses of chromatin architecture impact on reporter gene expression and chromatin immunoprecipitation experiments suggest that while Prmt5 activates transcription in a "classical" way in hematopoiesis, it rather functions as a scaffold protein that provides a proper chromatin conformation to promote endothelial gene expression during vascular morphogenesis.

## Results

### Prmt5 is required for blood cell formation

To characterize *prmt5* function, we generated a *prmt5* mutant by targeting the second exon of *prmt5* with the CRISPR/Cas9 system. A deletion of 28 nucleotides was obtained, leading to a premature stop codon before the catalytic domain of Prmt5 (Fig 1A). As a consequence, Prmt5, which was expressed ubiquitously in the trunk at 24 hours post fertilization (hpf), was no longer detected in the mutant (Fig 1B and 1C). In order to test whether Prmt5 regulates hematopoiesis in zebrafish, we took advantage of the transgenic line *Tg*(*gata2b*:*Gal4;UAS*:*lifeactGFP)* that labels Hematopoietic Stem Cells (HSCs) [28]. HSCs emerge from the ventral wall of the dorsal aorta (DA, Fig 1D and 1D'), before migrating into the Caudal Hematopoietic Tissue (CHT) (Fig 1D) where Hematopoietic Stem and Progenitor Cells (HSPCs) proliferate and undergo maturation [28]. Reminiscent of the situation in mice [16], the loss of *prmt5* led to an increased number of *gata2b*+ HSCs in 36 hpf mutant embryos as compared to wild type ones (Fig 1E–1G). In addition, we found that the relative expression of *runx1* or *cmyb*, which are specifically expressed in emerging HSCs, was increased in *prmt5* mutant embryos as compared to wild type embryos (Fig 1H). Of note, *prmt5* loss-of-function by *prmt5*-specific morpholino injection resulted in a similar increase of HSCs as compared to control morphants (S1A Fig). In order to further characterize the role of Prmt5 in HSC proliferation, we assessed the percentage of HSCs (gata2b+ cells) in mitosis by co-immunostaining for Phospho Histone H3 (P-H3). We found that, as in mouse [16], the increase of HSC number was associated with enhanced proliferation in *prmt5* morphants as compared to control embryos (10.15% *versus* 1.92% of gata2b+, P-H3+ cells) (S1B and S1C Fig). Altogether, these results suggest that Prmt5 regulates the number of emerging HSCs from the dorsal aorta. We next investigated whether blood cell formation was impaired in *prmt5* zebrafish mutant as described in mouse [16]. HSPCs give rise to different blood cell progenitors, such as lymphoid progenitors which colonize the thymus leading to T lymphopoiesis, but also erythroid or myeloid progenitors that will home in the CHT (Fig 1D) [29]. We found that at 5 days, the number of *gata2b*+ HSPCs in the thymus was significantly reduced in *prmt5* mutant as compared to wild-type (Fig 1I–1K). Similarly, imaging the caudal vascular plexus of *prmt5* morphant and control embryos at

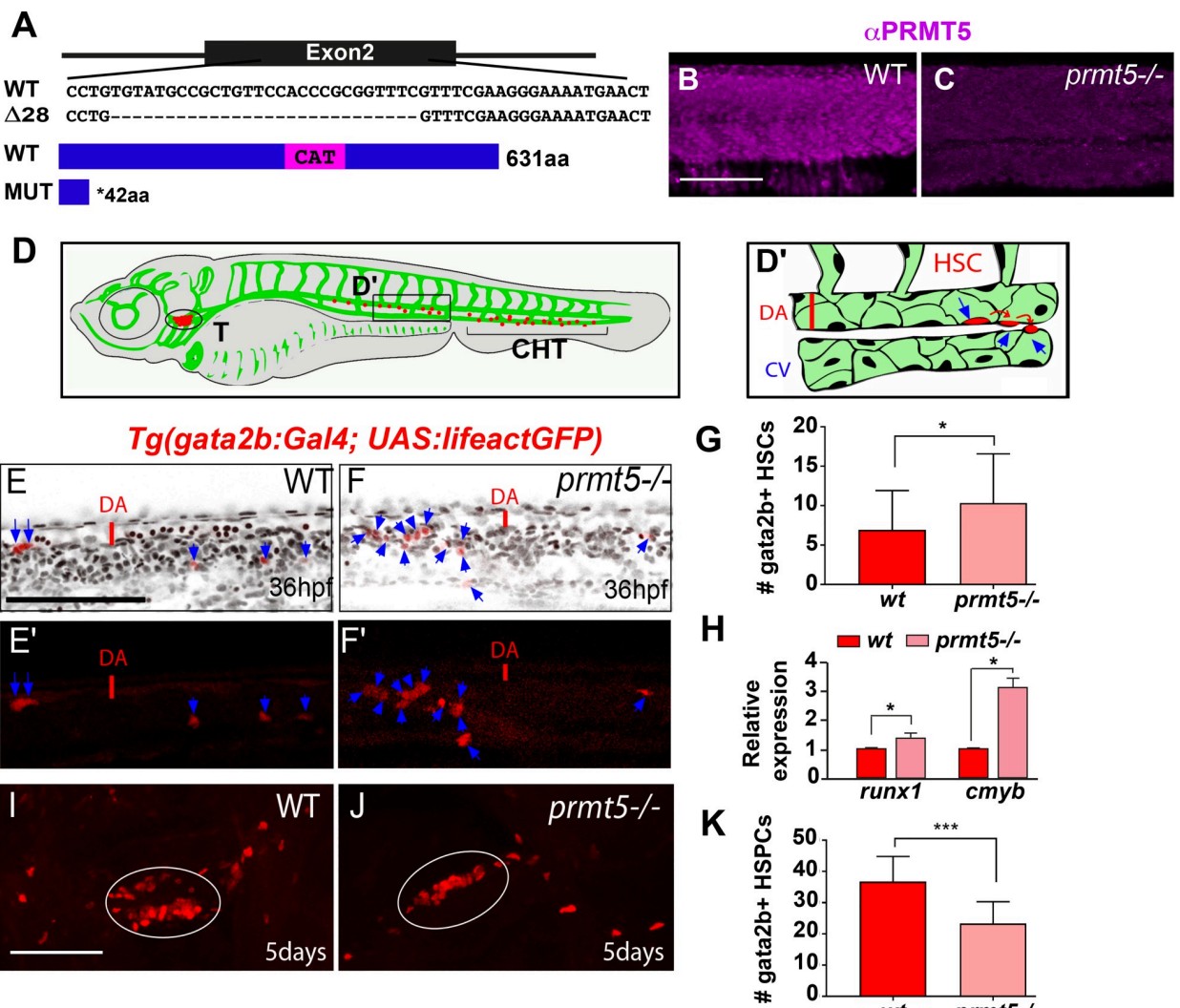

**Fig 1. Loss of *prmt5* affect HSCs and HSPCs production.** (A) Schematic representation of the sequence targeted by CRISPR/Cas9 leading to a 28 nucleotides deletion, and of wild type and truncated Prmt5 proteins. The catalytic domain "CAT" appears in magenta. (B, C) Confocal sections of immunostaining with anti-Prmt5 antibody of wild type and *prmt5* mutant embryos at 24 hpf. Scale bar 100 μm. (D) Schematic representation of vascular (green) and hematopoietic (red) systems in a zebrafish larva. Circle and bracket indicate the Thymus (T) and the Caudal Hematopoietic Tissue (CHT), respectively. (D') Close-up of the trunk vasculature where HSCs emerge from the ventral wall of the dorsal aorta (DA), bud and migrate. Red line represents the diameter of the dorsal aorta. Cardinal Vein (CV). (E-F') Confocal projections of transgenic *Tg* (*gata2b:Gal4; UAS:lifeactGFP*) embryos at 36 hpf showing gata2b+ cells in red and TO-PRO-3 (cell nuclei) in black. Blue arrows indicate HSCs labelled in red in wild type (E, E') and in *prmt5* mutant (F, F') embryos. Scale bar 100 μm. (G) Average number of HSCs enumerated per confocal stack in wild type and in *prmt5* mutant embryos at 36 hpf. Data are from 3 independent experiments with at least 6 individuals per experiment and a Mann-Whitney test was performed. (H) Relative mRNA expressions determined by RT-qPCR in the trunk of 36 hpf wild type and *prmt5* mutant embryos, from 3 independent experiments with at least 6 animals per condition. T-test was performed. (I, J) Confocal projections of wild type (I) and *prmt5* mutant (J) thymus from transgenic Tg(*gata2b:Gal4; UAS:lifeactGFP*) embryos at 5 days. Thymus are delimited by a white circle. Scale bar 100 μm. (K) Average number of HSPCs enumerated per confocal stack in wild type and *prmt5* mutant embryos at 5 days from 3 independent experiments with at least 5 individuals per analysis. T-test was performed. * P<0.05, ** P<0.01, ***P<0.001.

3 dpf showed a reduced number of *gata2b+* HSPCs in the CHT *of prmt5* morphants. Thus, as in mouse [16], Prmt5 seems required for HSPC maintenance in zebrafish (S1D–S1G Fig). Furthermore, analyzing the expression of the common lymphoid progenitor marker *rag1* and of the mature T cell marker *rag2* showed that both genes were downregulated in *prmt5* morphant as compared to control embryos (S1H and S1I Fig). In contrast, the expression of the erythrocyte marker *gata1* was increased and that of the myeloid marker *spi1/pu.1* was not affected

(S1J and S1K Fig). These data suggest that *prmt5* loss impairs lymphoid progenitors and T cell differentiation but promotes the production of erythrocyte without affecting myelopoiesis. In sum, our data support the conclusion that Prmt5 plays an important and conserved role for hematopoiesis in vertebrates.

## Prmt5 function in hematopoiesis depends on its methyltransferase activity

To further investigate Prmt5 mode of action in hematopoiesis, we tested whether its enzymatic activity was required in this process. Indeed, Prmt5 function has been shown to be generally due to its enzymatic activity, and since it is upregulated in many types of cancers, it constitutes a promising therapeutic target (for a review, [20]). Hence, intense studies have been undertaken to identify inhibitors of its methyltransferase activity and some of them, such as the EPZ015666 compound, are already tested in clinical trials [30].

To determine the impact of Prmt5 methyltransferase on the fate of gata2b+ cells in the thymus, we combined morpholino-based loss-of-function with rescue experiments using either a wild-type or a catalytic dead point mutant of human *prmt5* mRNA, all of which have been previously validated [16,23,31]. To first validate the effect of *prmt5* morpholino and the respective rescues, immunostainings were performed on 5 hpf embryos to detect either Prmt5, its obligate co-factor Mep50/Wdr77 or symmetric dimethylarginines (the generic product of arginine methylation by Prmt5 and some other Prmts) on control embryos, *prmt5* morphants and *prmt5* morphants injected with wild-type or with catalytic point mutant *prmt5* mRNA (Fig 2A–2L). As expected, Prmt5 expression was strongly decreased in *prmt5* morphant and was rescued after injection of *prmt5* wild type or point mutated mRNAs (Fig 2A–2D). Interestingly, Mep50/Wdr77 expression followed the same trend as Prmt5, supporting the finding that they regulate each other [32](Fig 2E–2H). Symmetric dimethylation levels slightly decreased in *prmt5* morphants as compared to control embryos and strongly increased upon injection of wild type *prmt5* mRNA but not of *prmt5* mutant mRNA (Fig 2I–2L). This confirms that *prmt5 wt* mRNA rescues efficiently Prmt5 enzymatic activity while *prmt5* mRNA mutated form is indeed coding a catalytic dead Prmt5 protein. Altogether, these data validate our approach as we observed the loss of Prmt5/Mep50 complex following *prmt5* morpholino injection, the efficient rescue of the complex and of its enzymatic activity by *prmt5 wt* mRNA, and the rescue of the complex but not of its enzymatic activity with *prmt5* catalytic mutant mRNA. Moreover, Prmt5 immunostaining in 24 hpf embryos validated the efficiency of *prmt5* loss-of-function and rescues at later stage (Fig 2M–2P). Hence, we used this approach to determine whether Prmt5 enzymatic activity was required for the formation of gata2b+ cells in the thymus at 3 dpf using the *Tg(gata2b:Gal4;UAS:lifeactGFP)* line (Fig 2Q–2T). We found that gata2b+ cell number was reduced in *prmt5* morphants as compared to control embryos and this effect was rescued upon injection of *prmt5 wt* mRNA but not of *prmt5* catalytic mutant mRNA (Fig 2Q–2T and 2X). These data underscore the conserved requirement of Prmt5 in hematopoiesis among species, as human *prmt5* mRNA could efficiently rescue Prmt5 function in *prmt5* morphants, as previously shown in myogenesis [23]. More importantly, they also show that Prmt5 enzymatic activity is required for gata2b+ cell formation in the thymus. To further support these findings, we assessed the impact of the well-established Prmt5 specific inhibitor (EPZ015666) on wild-type zebrafish embryos. EPZ015666 treatment efficiently inhibited Prmt5 activity *in vivo* as increasing concentrations of this drug led to a dose-dependent decrease of global symmetric dimethylarginines, as seen by western blot (Fig 2W). Interestingly, we found that EPZ015666 treatment also led to a decrease of gata2b+ cell number which was similar to the one observed in *prmt5* morphant (Fig 2U, 2V and 2X). Hence, our data show that Prmt5 enzymatic activity is required for its function in hematopoiesis.

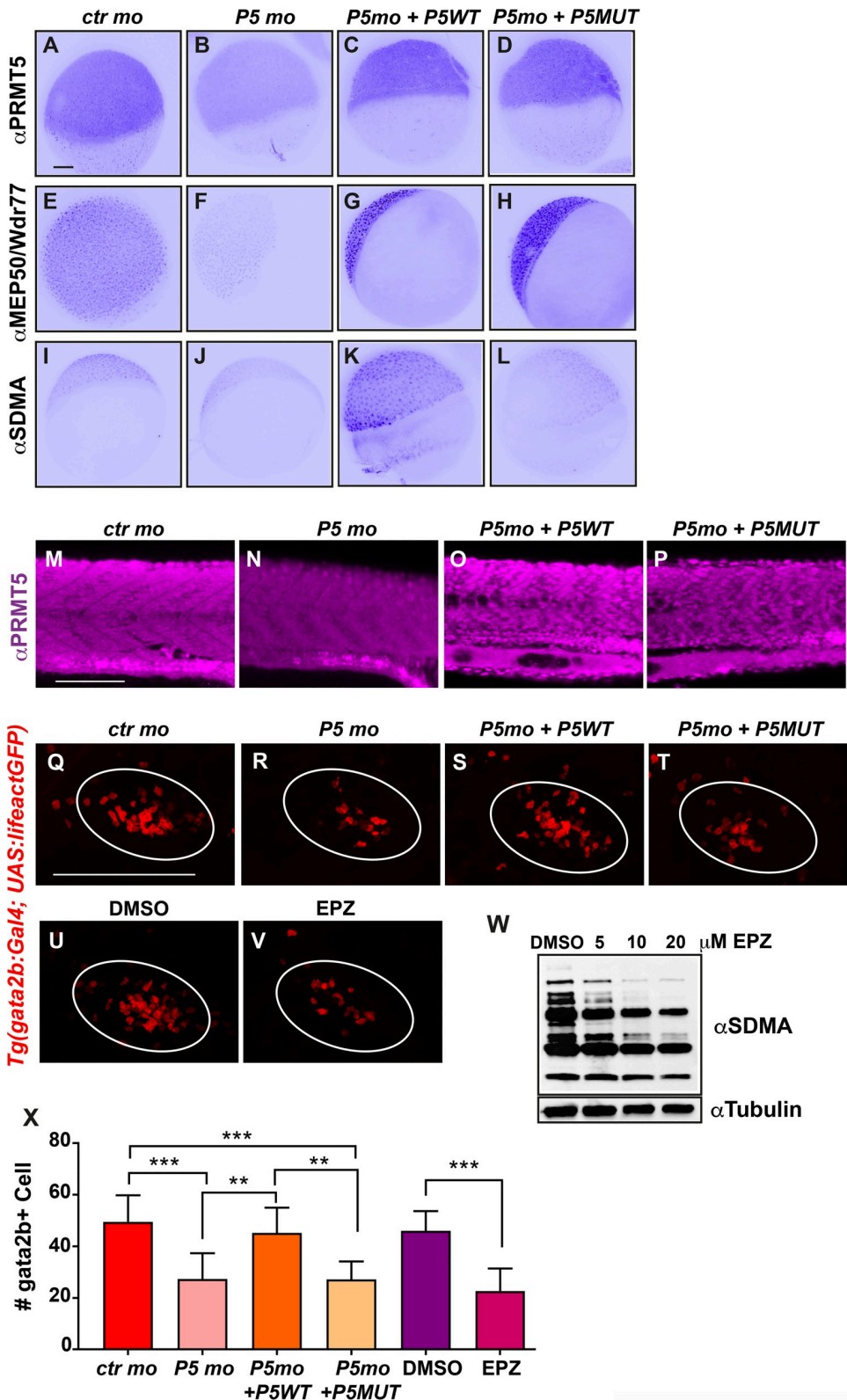

**Fig 2. PRMT5 methyltransferase activity is required for HSPC maintenance.** (A-D) PRMT5 immunostaining of 5 hpf embryos injected by control morpholino (A), or *prmt5* morpholino only (B) or in combination with *prmt5WT* mRNA (C) or with the catalytic mutant form *prmt5MUT* mRNA (D). (E-H) Immunostaining of embryos as in A-D

with an antibody raised against MEP50/Wdr77 PRMT5 obligate co-factor. (I-L) Immunostaining as in A-D but with an antibody recognizing SDMAs (symmetric Di-Methyl Arginine motifs) which are PRMT5 substrates. Scale bar, 100 μm. (M-P) Confocal projections of Prmt5 immunostaining of embryos injected by control morpholino (M), or *prmt5* morpholino only (N) or in combination with *prmt5WT* mRNA (O) or with the catalytic mutant form *prmt5MUT* mRNA (P) at 24 hpf. Scale bar 100 μm. (Q-T) Confocal projections of thymus rudiment from transgenic Tg(*gata2b*:*Gal4; UAS:lifeactGFP*) embryos at 3 days in the same conditions as in M-P. Thymus is delimited by a white circle. (U-V) Confocal projections of thymus rudiment from transgenic Tg(*gata2b*:*Gal4; UAS:lifeactGFP*) embryos treated with either DMSO (U) or 20 μM of PRMT5 inhibitor EPZ015666 (V) from 30 hpf to 3 dpf. Thymus is delimited by a white circle. Scale bar 100 μm. (W) Western blot of protein extracts from 24 hpf embryos treated either with DMSO or with increasing concentrations (5–20 μM) of EPZ015666, revealed with an antibody recognizing SDMAs. Reblotting with α tubulin antibody served as a loading control. (X) Average number of HSPCs enumerated at 3 days per confocal stack in injected embryos as in Q-V. Data are from 2 independent experiments with at least 3 individuals per analysis. T-test was performed. $^{**}P<0.01$. $^{***}P<0.001$.

## Prmt5 is required for vascular morphogenesis

As Prmt5 regulates zebrafish hematopoiesis, we next asked whether Prmt5 could also play a role during blood vessel formation, either during angiogenesis or vasculogenesis. First, we analyzed the expression and localization of Prmt5 by immunostaining in *Tg(fli1a:eGFP)* transgenic embryos, in which endothelial cells can be visualized with *egfp* [4]. We observed that Prmt5 was expressed in early endothelial cells at 14 somite-stage (S2A–S2A" Fig). At 24 hpf, Prmt5 was expressed in endothelial cells of the dorsal aorta (DA) and of the cardinal vein (CV) (S2B, S2B' and S2D Fig) and was also detected in Intersegmental Vessels (ISVs) sprouting from the DA, in the tip cell (leading the sprout) and the stalk cell (S2C, S2C' and S2D Fig). Single cell RNA-sequencing data [33] supported the expression of *prmt5*, but also of its obligate cofactor *wdr77/mep50*, in endothelial cells from 10 hpf to 24 hpf with a slight decrease in the later stages (S2E Fig). We then analyzed whether blood vessel formation was affected in transgenic *Tg(fli1a:eGFP); prmt5* mutants at 28 hpf. We found that *prmt5* mutants had a defect for sprouting ISV: some of them did not reach the dorsal part of the trunk and failed to connect with other ISVs to form the Dorsal Longitudinal Anastomotic Vessel (DLAV) (Fig 3A–3C). This defect was associated with a significant reduction of ISV length (Fig 3D) but with no impact on endothelial cell number (Fig 3E). Importantly, *prmt5* morphants reproduced the phenotype observed in *prmt5* mutants *i.e.* a reduced ISV length without any reduction of the cell number per ISV (Fig 3F–3I), further validating the specificity of the morpholinos. The observed size reduction of ISVs is thus most likely the result of an elongation issue rather than a proliferation defect. This effect was not due to a developmental delay at early stage, as transgenic mutants imaged at 48 hpf do present a formed DLAV, but also an increased number of disconnected ISVs (S3A–S3C Fig), likely as a result of impaired vessel elongation. Mutant embryos at 28 hpf also displayed a reduced DA diameter as compared to the control (Figs 3B and 2C, close-ups), suggesting that lumen formation was perturbed. To confirm this result, we made use of the Notch reporter line *Tg(TP1bglob:VenusPEST)$^{s940}$* in which only the dorsal aorta cells express the transgene while the cardinal vein endothelial cells do not [34,35]. In this context, the area occupied by the dorsal aorta in *prmt5* morphant embryos was significantly reduced as compared to control embryos (Fig 3J–3L). Similarly, at 48 hpf the area occupied by ISVs in mutant embryos was significantly decreased as compared to control embryos (S3D Fig), thus revealing lumen formation defect at later stages.

To get a better insight into the impact of Prmt5 on the dynamics of vascular system formation, we performed time-lapse analyses with the *Tg(Fli1a:eGFP)* line in control and *prmt5* morphant embryos. Time-lapse confocal movies were carried out from 28 hpf to 38 hpf to follow ISV elongation of ISVs and lumen formation. As compared to control, *prmt5* morphants showed impaired formation of ISV lumen and DLAV (Fig 3M and 3N). Indeed, in *prmt5*

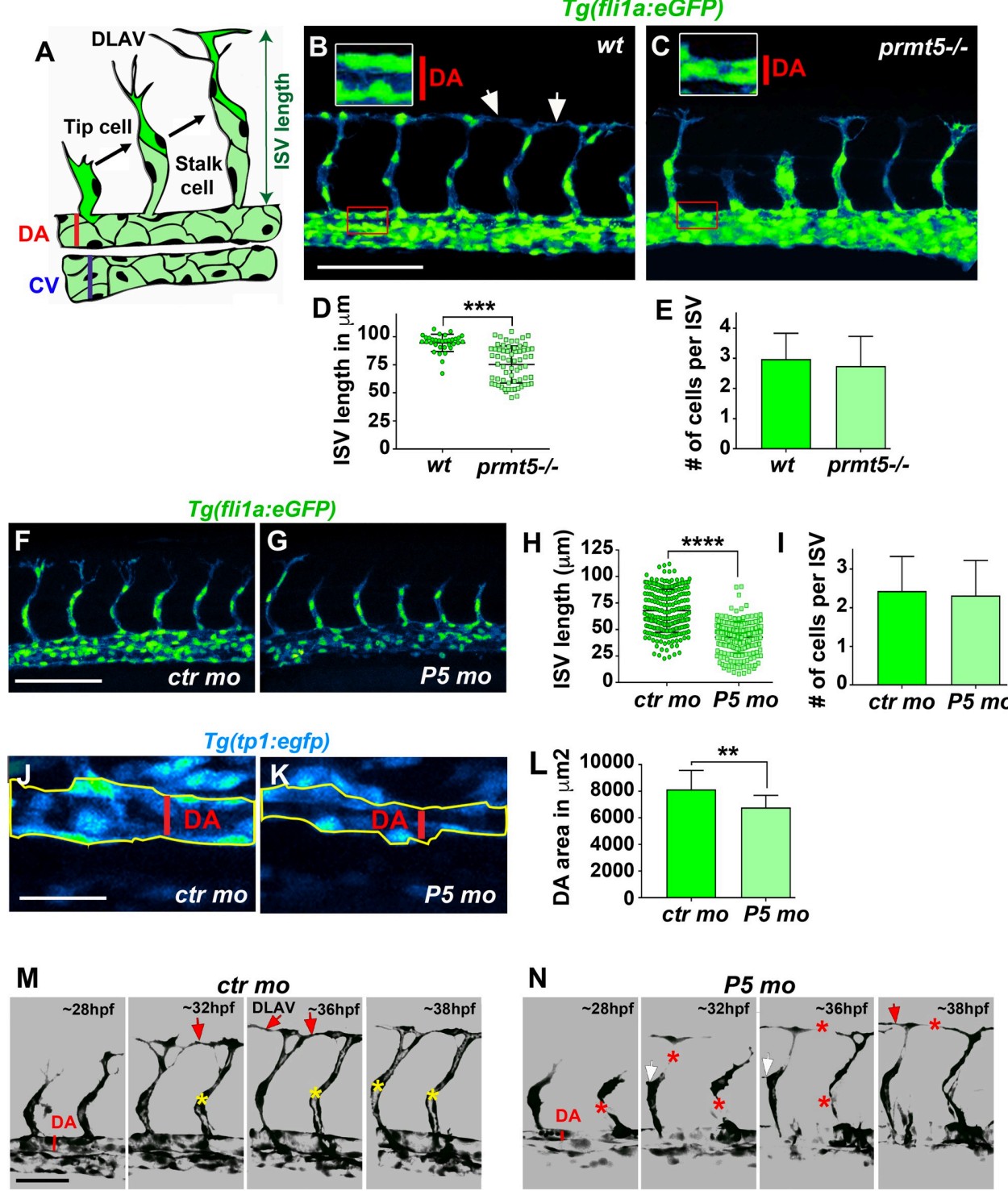

**Fig 3. Loss of *prmt5* impairs blood vessel formation.** (A) Schematic representation of the trunk vasculature with Intersegmental Vessels (ISV) sprouting from the dorsal aorta (DA). The tip cell leads the cell migration and the stalk cell maintains the connection with the DA. CV: cardinal vein. (B, C) Confocal projections of transgenic *Tg(fli1a:GFP)^y1* wild-type (B) and *prmt5* mutant (C) embryos at 28 hpf. Red rectangles delimit where DA close ups were made. White rectangles delimit the higher magnification (x2) of the DA with red lines indicating the dorsal aorta diameters. White arrows indicate the connection point between two ISVs to form the Dorsal Longitudinal Anastomotic Vessel (DLAV). Scale bar 100 μm. (D, E) Average ISV length in μm (D) and average number of endothelial cells per intersegmental vessel (E) in control and in *prmt5* mutant embryos from 3 independent experiments with at least 3 animals per condition. T-test and Mann Whitney test were performed, respectively. ** P<0.01,

***P<0.001. (F, G) Confocal projections of transgenic Tg(*fli1a:GFP*)[y1] embryos injected by either control morpholino (F) or *prmt5* morpholino (G) at 28 hpf. Scale bar 100 μm. (H, I) Average ISV length in μm (H) and average number of endothelial cells per ISV (I) in control and *prmt5* morphant embryos, from 4 independent experiments with at least 5 animals per condition. Mann-Whitney test were performed. **** P<0.0001. (J, K) Confocal projections of control morphant (J) and *prmt5* morphant (K) transgenic *Tg(TP1bglob:VenusPEST)*[s940] embryos labelling cells of the DA at 28 hpf. Yellow lines delimit the measured area occupied by the DA. Scale bar 25 μm. (L) Average area occupied by the DA in μm$^2$ in control and *prmt5* morpholino injected embryos from 2 independent experiments with at least 8 animals per condition. T-test was performed. (M, N) Still images from movies of control (M) and *prmt5* morphant (N) *Tg(fli1a:GFP)*[y1] transgenic embryos from 28 to 38 hpf. Red asterisks label missing connections between tip and stalk cells as well as missing connections between tip cells that should lead to DLAV formation. Red arrows point to connecting ISVs leading to DLAV formation. White arrows indicate supernumerary sprouts. Yellow asterisks label the lumen of ISVs. Scale bar 50 μm.

morphants tip cells failed to stay connected with the stalk cells and to contact other tip cells to form the DLAV. Moreover, supernumerary connections were detected in *prmt5* loss-of-function. Of note, re-expressing *prmt5* specifically in endothelial cells in *prmt5* morphants was sufficient to rescue partially ISV length (S3E–S3I Fig) as well as *cdh5* expression, a gene downregulated in *prmt5* mutant (see below), supporting a cell autonomous role for Prmt5 in endothelial cells. Altogether, these data suggest a central role of Prmt5 in vascular morphogenesis.

## Prmt5 methyltransferase activity is not required for vascular morphogenesis

Next, we asked whether Prmt5 methyltransferase activity was also required for vascular morphogenesis as observed for blood cell formation. To this end, transgenic *Tg(fli1a:eGFP) prmt5* mutant embryos were injected with either wild-type *prmt5* mRNA or with *prmt5* mRNA mutated in the catalytic site (Fig 4A–4D), or were treated with Prmt5 inhibitor EPZ015666 (Fig 4E and 4F). Surprisingly, both mRNAs were able to restore ISV elongation, as indicated by the average ISV length in injected mutant embryos as compared to non-injected mutants (Fig 4G). However, the average length of ISVs in *prmt5 wild-type*-injected mutants was even longer than intersegmental vessels of wild-type embryos, while the average length in *prmt5 catalytic dead*-injected mutants was significantly superior to non-injected mutants but similar as control embryos (Fig 4G). Moreover, EPZ015666-treated embryos did not show any significant reduction of ISV length compared to control embryos, further supporting the idea that Prmt5 enzymatic activity is not required for ISV elongation (Fig 4G). Interestingly, no difference could be detected in the cell number per ISV in the different contexts, thus ruling out the possibility that Prmt5 regulates ISV cell proliferation (Fig 4H). Therefore, in contrast to blood cell formation, angiogenesis seems largely independent of Prmt5 methyltransferase activity.

We next wondered whether Prmt5 regulates blood vessel formation through the control of the master regulator Etv2, ETS transcription factors and adhesion proteins, all known to be involved in this process [8–10,36,37]. Analyzing single cell RNA-sequencing data [33] allowed us to confirm that *etv2*, *fli1a*, *fli1b*, *cdh5*, *agtr2* and *esama* are expressed in endothelial cells from 10 hpf and showed that their expression increases from 14 to 24 hpf (S2E Fig). In addition, our previous RNA-seq results show that these 5 genes are essentially expressed in endothelial cells and not in other cell types in 24 hpf embryos (S1 Table)[38]. To test whether Prmt5 could regulate the expression of these genes, RT-qPCR experiments were performed on *prmt5* mutant embryos and on their wild-type counterparts (Fig 4I–4K). As expected, *prmt5* expression was reduced in the mutant but *etv2* expression was not affected (Fig 4I and 4J). However, the expression of ETS transcription factors (*fli1a*, *fli1b*) and adhesion proteins *(cdh5*, *agtr2* and *esama)*, all putative Etv2 target genes [6,7], was significantly reduced in *prmt5* mutant (Fig 4K). Of note, the reduction of *fli1a* and *cdh5* expression as detected in *prmt5* mutant by RT-qPCR was confirmed by *in situ* hybridization in the hemogenic endothelium (S3J–S3S Fig). That *etv2* expression was unaffected by the loss of *prmt5* whereas its targets

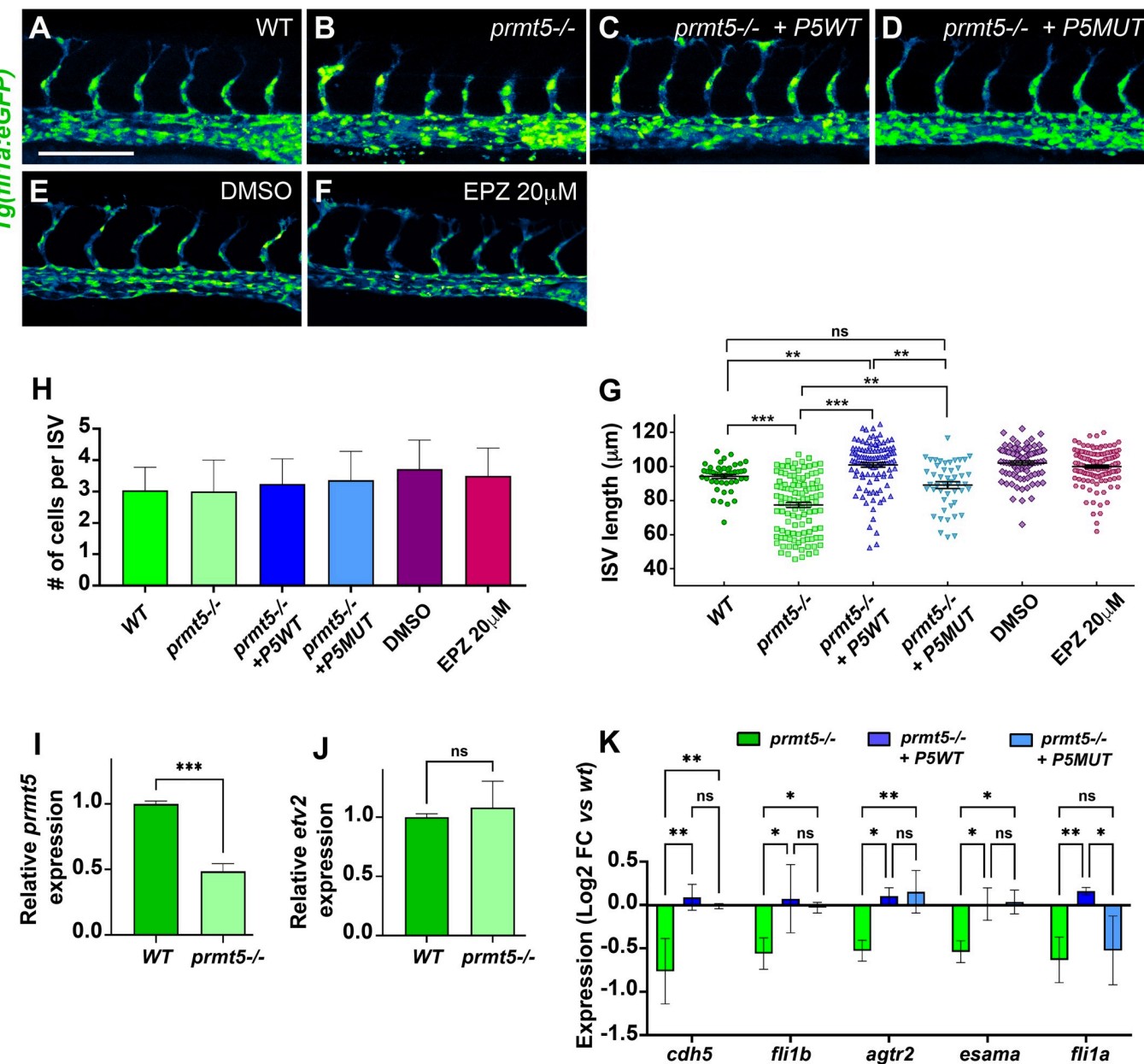

**Fig 4. Prmt5 requirement for vascular morphogenesis is independent of its methyltransferase activity.** (A-D) Confocal projections of transgenic *Tg(fli1a: GFP)*[y1] embryos at 28 hpf. Wild type embryo is on the top left panel (A), *prmt5* mutant embryos were not injected (B) or injected with either *prmt5WT* mRNA (C) or the mutant form *prmt5MUT* mRNA (D). Scale bar 100 μm. (E, F) Confocal projections of transgenic *Tg(fli1a:GFP)y1* embryos treated with either DMSO (E) or EPZ 20 μM (F) from 9 hpf to 28 hpf. (G, H) Average ISVs length in μm (G) and average number of endothelial cells per ISVs (H) of embryos. Data were from 3 independent experiments with at least 3 animals per condition. Kruskal-Wallis test (G) and One-way ANOVA (H) were performed. * $P<0.05$, ** $P<0.01$, *** $P<0.001$, **** $P<0.0001$. (I, J) Relative expression of *prmt5* (I) or *etv2* (J) mRNA by RT-qPCR on 28 hpf wild type and *prmt5* mutant embryos from 3 independent experiments with at least 6 animals per condition. T-test was performed. ***$P<0.001$. (K) Log2 fold change expression relative to *wt* embryos of *cdh5*, *fli1b*, *agtr2*, *esama* and *fli1a* mRNAs by RT-qPCR on 28 hpf *prmt5* mutant embryos not injected or injected by either *prmt5WT or prmt5MUT* mRNAs, from at least 2 independent experiments with a minimum of 6 animals per condition. Two-way ANOVA was performed. * $P<0.05$. **$P<0.01$.

were down-regulated suggests that Prmt5 is required to promote Etv2 target gene expression. Interestingly, RT-qPCR experiments on the tails of *prmt5* mutant embryos injected with either wild-type *prmt5* mRNA or *prmt5* mRNA mutated in the catalytic activity revealed that both mRNAs were able to restore the expression of all Etv2 target genes, except *fli1a* whose

expression was only rescued by *prmt5* wild-type (Fig 4K). In sum, these results indicate that Prmt5 controls blood vessel formation by regulating Etv2 target genes mostly independently of its methyltransferase activity.

## Prmt5 might regulate gene expression by shaping correct chromatin conformation in endothelial cells

As Prmt5 methyltransferase activity seems dispensable for gene activation during vascular morphogenesis, we speculated that Prmt5 could act as a scaffold protein in complexes mediating transcription and chromatin looping. Indeed, Prmt5 has been proposed to promote enhancer-promoter looping at the PPARγ2 locus and more broadly to facilitate chromatin connection in adipocytes, *via* the recruitment of the Mediator subunit MED1 and the SWI/SNF chromatin remodeling complex subunit Brg1 [39]. Thus, we decided to inspect the chromatin architecture of the flanking region of Prmt5-regulated genes using ATAC-seq data from zebrafish endothelial cells [38]. Doing so, we found putative endothelial-specific enhancers in *agtr2*, *cdh5*, *esama*, *fli1a and fli1b*. Interestingly *cdh5*, *esama* and *fli1b* loci presented enhancers at relatively long distances (>10kb) from the transcriptional start site (TSS) suggesting that their expression could be regulated by Prmt5 through the formation of chromatin loop (S4 Fig). To explore this hypothesis, we performed chromatin immunoprecipitation experiments to test whether Prmt5, MED1 and/or Brg1 were recruited to the promoters as well as to the putative distant endothelial enhancers of these three genes. Of note, similar to Prmt5, MED1 and Brg1 were expressed in endothelial cells at 24 hpf [38](S1 Table). Importantly, ChIP experiments showed that Prmt5, MED1 and Brg1 were all binding to *cdh5* promoter and to its putative enhancer, though at different levels (2–3 fold enrichment for Prmt5 and MED1; 4–8 fold for Brg1), this variation being probably due to antibody efficiency differences between these factors (Fig 5A and 5B). Also, since *cdh5* expression is restricted to endothelial cells, which correspond to a small proportion of cells within whole embryos, it was not surprising to observe a relatively weak enrichment in ChIP assays. Testing *esama* promoter and its four putative enhancers by ChIP showed a recruitment of all three cofactors on the promoter and on Enhancer 3 (E3), but no recruitment on the three other enhancers (E1, E2 and E4) (Fig 5C and 5D). Furthermore, we observed a similar scenario for *fli1b*, *i.e* an efficient binding of Prmt5, MED1 and Brg1 to the promoter and to enhancer 4 (E4), but no recruitment of these proteins on enhancer 5 (E5) (Fig 5E and 5F). Hence, our ChIP data indicate that Prmt5, MED1 and Brg1 are recruited altogether on promoters and on selected distal enhancers of *cdh5*, *esama* and *fli1b*. These results suggest that the three factors could participate to a chromatin loop between the promoter and the corresponding enhancer of these genes to activate their expression.

In order to gain further insight into the role of Prmt5 in supporting chromatin looping to induce endothelial gene expression, we made use of two reporter transgenes in which the *cdh5* enhancer, which we found to be bound by Prmt5, MED1 and Brg1, is either in its natural context (*i.e.* largely separated from its promoter) or inserted just before its promoter. First, we analyzed the expression of the *TgBAC(cdh5:GAL4FF) Gal4* transgene that contains the sequence of an optimized version of Gal4VP16 (*GAL4FF*) inserted in the endogenous chromatin context of *cdh5*, *i.e.* at the TSS of *cdh5*, between its promoter and putative enhancer distant of ~20kb, as defined by the presence of two ATAC-seq positive regions (Figs 5G and S4)[38,40]. Therefore, in double transgenic individuals *TgBAC(cdh5:GAL4FF);Tg(UAS:GFP)*, the level of GFP fluorescence intensity correlates with endogenous *cdh5* expression. Comparing the level of fluorescence intensity in *TgBAC(cdh5:GAL4FF);Tg(UAS:GFP)* transgenic line in control condition and in the context of *prmt5* knock-down, we observed a strong reduction of GFP

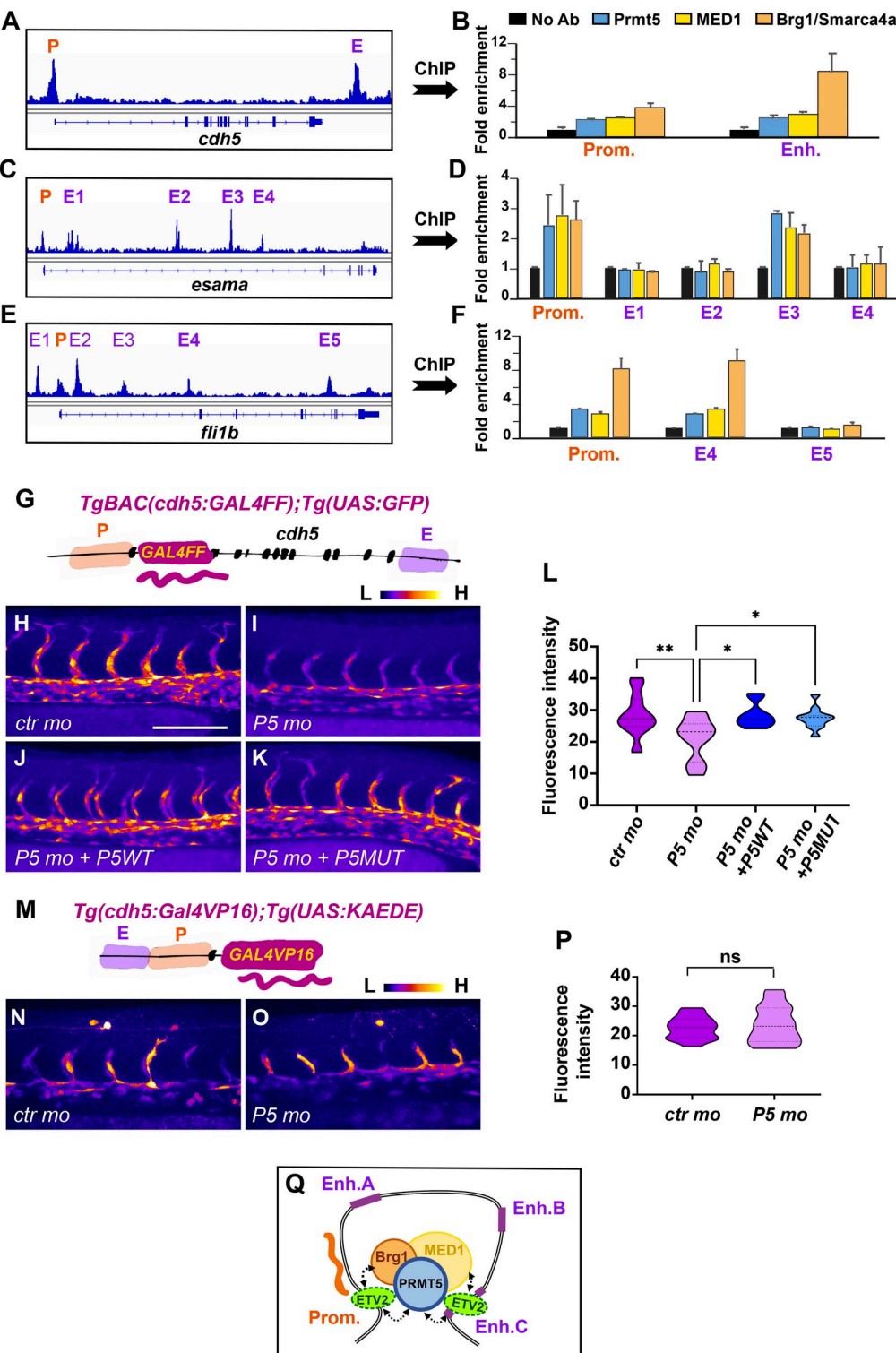

**Fig 5. Prmt5 promotes chromatin looping.** (A-F) ChIP experiments with the indicated antibodies on *cdh5* (A, B), *esama* (C, D) and *Fli1b* (E, F) regulatory sequences. P, promoter; E, enhancer. Samples were analyzed in duplicates from 2 experiments. Fold enrichment was calculated relative to the input and to a negative region on the genome and then relative to the mock ChIP (No antibody). (G) Schematic representation of the transgene *TgBAC(cdh5:GAL4FF)* containing two putative cis-regulatory elements, a promoter region (P) and an enhancer (E), separated by ~20kb with the *GAL4FF* reporter gene inserted

at the TSS of *cdh5*. (H-K) Confocal projections of transgenic *TgBAC(cdh5:GAL4FF);*Tg(*UAS:GFP*) embryos at 28 hpf. Control morphant is on the top left panel (H), *prmt5* morphant embryos were not injected (I) or injected by either *prmt5WT* mRNA (J) or the catalytic mutant form *prmt5MUT* (K) mRNA. The fluorescent intensity is colored-coded, from the Low intensity (L) in black to High intensity (H) in white. Scale bar 100 μm. (L) Average GFP fluorescence intensity per confocal projection for control, *prmt5* morphant embryos not injected or injected by *prmt5WT* mRNA or *prmt5 MUT*, from 3 independent experiments with at least 3 animals per condition. One-way ANOVA was performed. $^*P<0.05$, $^{**}P<0.01$. (M) Schematic representation of the transgene Tg(*cdh5:GAL4VP16*) containing the two putative *cis*-regulatory elements next to each other (E and P), upstream of *GAL4VP16* reporter gene. (N, O) Confocal projection of transgenic Tg(*cdh5:GAL4VP16*);Tg(*UAS:KAEDE*) embryos at 26 hpf injected with either a control morpholino (N) or *a prmt5* morpholino (O). The fluorescence intensity is color- coded, from the Low intensity (L) in black to High intensity (H) in white as in panels (H-F). (P)- Average KAEDE fluorescence intensity for control and for *prmt5* morphant embryos, from 3 independent experiments with at least 5 animals per condition. T-test was performed. (Q) Proposed model to depict the function of Prmt5 in zebrafish endothelial cells. The transcription factor Etv2 bound on promoters and selective enhancers of endothelial specific genes, could recruit a complex including Prmt5, Brg1 and the mediator complex which favors the formation of a chromatin loop thereby facilitating the transcription of these genes. Dashed lines indicate potential interactions for the recruitments of Prmt5, Brg1 and/or the mediator complex by Etv2.

fluorescence intensity in *prmt5* morphants (Fig 5H, 5I and 5L), thus confirming that Prmt5 is required for *cdh5* expression in an endogenous context. Interestingly, injecting either wild-type *prmt5* mRNA or mutated in Prmt5 catalytic activity in *prmt5* morphants rescued efficiently *cdh5* expression and restored GFP fluorescence intensity (Fig 5J–5L). These results further support the conclusion that Prmt5 directly regulates *cdh5* transcription in an enzymatic-independent manner. Second, we analyzed the expression of the *Tg(cdh5:Gal4VP16)* transgenic line in which *cdh5* promoter and putative enhancer were cloned next to each other, both upstream of the *Gal4VP16* coding sequence (Fig 5M). In double transgenic embryos *Tg(cdh5: Gal4VP16); Tg(UAS:KAEDE)*, the fluorescence intensity of the protein KAEDE is an artificial read out of *cdh5* transcription for which chromatin looping is not required. In these double transgenic embryos, the fluorescent protein KAEDE was expressed in blood vessels (Fig 5N), validating that the putative enhancer and the promoter region of *cdh5* are sufficient to drive specific gene expression in endothelial cells. Strikingly, *prmt5* morpholino injection had no effect on the level of KAEDE fluorescence intensity as compared to control morphants (Fig 5N–5P). Hence, it appears that tethering *cdh5* enhancer to the vicinity of its promoter is sufficient to bypass the requirement for Prmt5 to activate it.

Collectively, our chromatin immunoprecipitation experiments and our analyses of chromatin architecture impact on *cdh5* reporter gene expression suggest that Prmt5 regulates transcription by acting as a scaffold protein that facilitates chromatin looping of certain genes involved in vascular morphogenesis (Fig 5Q).

## Discussion

Here we demonstrate a role for Prmt5 in both hematopoiesis and blood vessel formation in zebrafish. We found that, as in mouse [16], Prmt5 plays an important role in zebrafish hematopoiesis by controlling HSC emergence and HSPC expansion and that this function depends on Prmt5 enzymatic activity. We also describe for the first time the involvement of Prmt5 in vascular morphogenesis, whereby it controls the expression of key ETS transcription factors and adhesion molecules, and we showed that this function is in good part independent of its methyltransferase activity. Analyses based on reporter gene expression and chromatin immunoprecipitation experiments led us to propose that Prmt5 regulates transcription by acting as a scaffold protein that facilitates chromatin looping to promote the expression of genes involved in vascular morphogenesis.

Our finding that Prmt5 controls the expression of ETS transcription factors and adhesion proteins in endothelial cells is further supported by the fact that *prmt5* loss-of-function

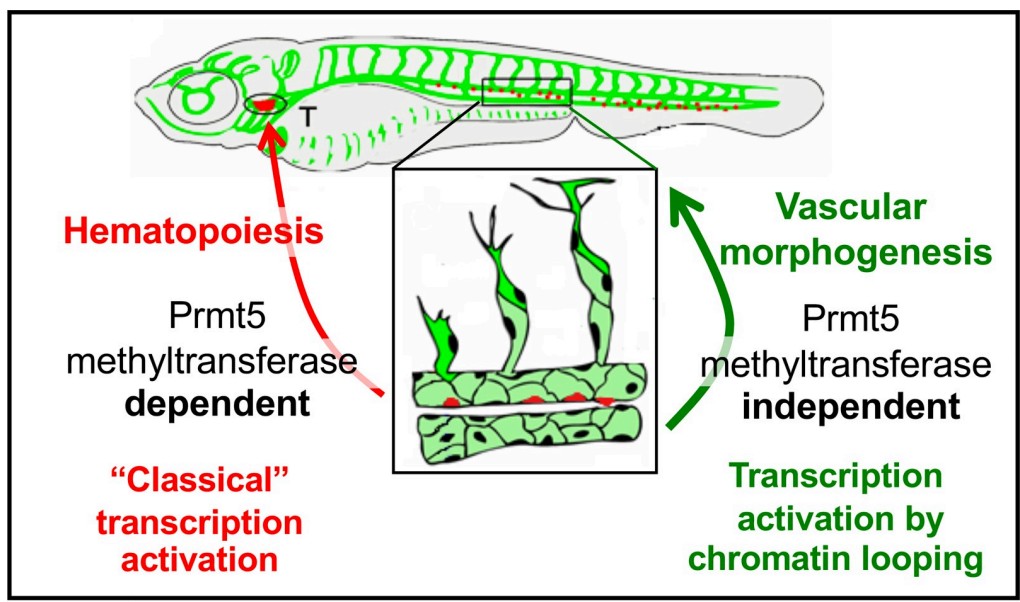

**Fig 6. Schematic representation of the two distinct roles of Prmt5 during the formation of hematopoietic lineages and blood vessels, relying or not on its methyltransferase activity, respectively.**

partially phenocopied loss-of-function(s) of these genes. Indeed, it was shown that knocking-down individual ETS proteins had limited effect on sprout formation, while the combination of morpholinos against both *fli1a*, *fli1b* and *ets1* led to a decreased number of vessel sprouts at 24 hpf [37]. Moreover, the disconnected stalk and tip cells and delayed formation of DLAV that we observed in *prmt5* mutant phenocopied the loss-of-function of both *cdh5* and *esama* [8,9]. However, *cdh5* loss had no effect on HSC emergence or HSPC expansion [41], suggesting that Prmt5 acts on different set of genes in endothelial cells and in emerging HSCs. In agreement with this hypothesis, Tan et al. proposed that Prmt5 plays a critical role in HSC quiescence through the splicing of genes involved in DNA repair [27]. Consistent with our findings, this study also showed that Prmt5 methyltransferase activity was required for controlling HSC quiescence. Importantly though, our data revealed that the methyltransferase activity of Prmt5 is largely dispensable in endothelial cells, reinforcing the idea that Prmt5 regulates transcription by different mechanisms in these two processes (Fig 6).

Prmt5 has been shown to facilitate ATP-dependent chromatin remodeling to promote gene expression in skeletal muscles and during adipocyte differentiation [24,31,39,42]. Here, we propose that Prmt5 could also be essential for proper chromatin looping in endothelial cells. Indeed we found that Prmt5 influences gene expression in contexts supporting chromatin looping (*e.g. cdh5* and *TgBAC(cdh5:GAL4FF))*, while it is dispensable for gene expression when enhancer and promoter regions are artificially associated (*e.g. Tg(cdh5:Gal4VP16))*. Moreover, our ChIP experiments revealed the binding of Prmt5 but also of MED1 and Brg1 to the promoters and to specific distal enhancers of *cdh5*, *esama* and *fli1b*, which are regulated by Prmt5 in an enzymatic-independent way. MED1 or Brg1 have been shown to participate in the formation of chromatin loop between promoter and distant enhancer in several cases [43–47] and notably at the *PPARγ2* locus in mouse adipocyte cell culture, where they act together with Prmt5 [39]. This implies that Prmt5 could interact with Brg1 and MED1 to induce promoter-enhancer looping in endothelial cells *in vivo*. Interestingly, *brg1* mutant mouse embryos display an anemia coupled to vascular defects in the yolk sac, characterized by thin vessels and

supernumerary sprouts [13], which is reminiscent to our present findings in zebrafish with *prmt5* mutant. In addition, it has been proposed that the Mediator complex regulates endothelial cell differentiation [48]. It is thus tempting to speculate that Prmt5, Brg1 and MED1 act together to mediate chromatin looping and gene activation in endothelial cells (Fig 5Q).

The presence of Prmt5 and Brg1 at promoter regions of *PPARγ2* or *myogenin* was associated with dimethylated H3R8 (histone 3 arginine 8) and *prmt5* knock-down led to a reduction of both histone methylation and chromatin loop formation [42,49]. However, these studies did not assess whether the ability of Prmt5 to facilitate chromatin looping at these loci was functionally linked to its methyltransferase activity. Our data suggest that chromatin looping favored by Prmt5 does not require its methyltransferase activity. Indeed, rescue experiments demonstrated that Prmt5 was able to restore gene expression independently of its enzymatic activity, with the exception of *fli1a* expression. Since *fli1a* putative enhancers are located at proximity from its promoter, chromatin looping might not be required for *fli1a* expression and Prmt5 might essentially act here through its methyltransferase activity. Yet chromatin looping might not be the only enzymatic-independent mode of action of Prmt. Indeed, we did not identify distant enhancers for *atgr2*, whose expression was rescued by a catalytic-dead Prmt5. However, it is also possible that *atgr2* distal enhancers were no clearly apparent or were located too distally to be identified in the ATAC-seq data. Hence, depending on the context and the target genes, Prmt5 could modulate gene expression in endothelial cell through promotion of chromatin interaction and/or *via* histones/proteins modification. Still, in light of our results from rescue experiments and drug inhibitor treatments, the most critical functions of Prmt5 in vascular morphogenesis *in vivo* appear to be independent of its enzymatic activity.

Which transcription factor mediates Prmt5 recruitment to its target genes in endothelial cells remains unknown. ChIP-seq data available in mouse revealed that some flanking regions of orthologues of the Prmt5 target genes *cdh5*, *esama* and *fli1b* are bound by Etv2 [6]. Moreover, similar to *prmt5* mutant, *etv2* zebrafish mutant display abnormal vasculature at 48 hpf characterized by defective lumen formation, defective vessel extension and aberrant connections [36,37]. Hence, Etv2 could be involved in the recruitments of Prmt5 to *cis* regulatory regions of endothelial genes. However, we could not confirm this hypothesis as none of the commercially available Etv2 antibody was functional in our hands. Another crucial player of blood vessel formation is the transcription factor Npas4l, which is expressed during late gastrulation and regulates *etv2* expression [50]. Npas4l ChIP-seq data analysis revealed the binding of this transcription factor to several *cis*-regulatory regions of Prmt5 target genes identified in the present work. Hence, Npas4l could participate to the recruitment of Prmt5 to endothelial genes. Even though highly challenging technically due to the low amounts of endothelial cells in fish embryo, Cut&Run experiments in sorted endothelial cells against afore mentioned factors or any other factor of interest, combined with HiC data and the corresponding RNA-seq/ATAC-seq experiments in wild-type or mutant conditions for *Prmt5* would certainly help to identify Prmt5 target genes and its global impact on chromatin organization.

Finally, it is worth mentioning that Prmt5 has been proposed as a therapeutic target in many diseases, including cancer [12]. In light of our result, targeting Prmt5 might also reduce tumor growth indirectly by inhibiting tumor angiogenesis. Along that line, it was recently shown in a zebrafish xenotransplantation model that Etv2 and Fli1b are required for tumor angiogenesis [51]. Several Prmt5 inhibitors have been discovered in the past decade and some have been tested in clinical trials for the treatment of tumors (reviewed in [52]). However, these compounds were developed and validated for their ability to inhibit Prmt5 enzymatic activity [53]. Yet, we show here that Prmt5 acts at least in part, independently of its methyltransferase activity to regulate vascular morphogenesis and it is possible that enzymatic-independent function of Prmt5 controls other physiological and pathological situations. Hence,

our data shed light on a mechanism of action of Prmt5 that will be insensitive to the afore mentioned enzymatic inhibitors and calls forth the design of alternative drugs *i.e.* specific inhibitors of the interaction between Prmt5 and Etv2, MED1 or Brg1 for example.

In conclusion, our study highlights different modes of regulation of gene expression by Prmt5 in endothelial cells *in vivo* and underscores the importance of its enzymatic-independent function in chromatin looping. This non-canonical function of Prmt5 may have a more pervasive role than previously thought and certainly deserves more attention in the future.

## Materials and methods

### Ethics statement

Fish were handled in a facility certified by the French Ministry of Agriculture (approval number A3155510). The project has received an agreement number APAFIS#7124–20161 00517263944 v3. Anesthesia and euthanasia procedures were performed in Tricaine Methanesulfonate (MS222) solutions as recommended for zebrafish (0.16 mg/ml for anesthesia, 0.30 mg/ml for euthanasia). All efforts were made to minimize the number of animals used and their suffering, in accordance with the guidelines from the European directive on the protection of animals used for scientific purposes (2010/63/UE) and the guiding principles from the French Decret 2013–118.

### Zebrafish care and maintenance

Embryos were raised and staged according to standard protocols and the Recommended Guidelines for Zebrafish Husbandry Conditions [54,55]. The establishment and characterization of *Tg(gata2b:Gal4;UAS:lifeactGFP)*, *Tg(fli1a:eGFP)*, *Tg(TP1bglob:VenusPEST)s940*, *TgBAC(cdh5:GAL4FF);Tg(UAS:GFP)*, *Tg(UAS:KAEDE)* have been described elsewhere [4,28,34,40,56]. Lines generated in this study are described below. Embryos were fixed overnight at 4˚C in BT-FIX, after which they were immediately processed or dehydrated and stored at −20˚C until use.

### Plasmid construction

To construct the transgene *Tg(cdh5:GAL4VP16)*, we cloned the putative *cdh5* promoter (*cdh5*P) and enhancer (*cdh5*E) elements into pme_mcs and p5E_GGWDest+ (Addgene #49319) [57,58] using XhoI, EcoRI and BsaI to generate pme_*cdh5*P and p5E_*cdh5*E, respectively. The Gal4VP16 sequence from pme_Gal4VP16 [58] was then introduced downstream of *cdh5*P into pme_*cdh5*P using BamH1 and SpeI. A multisite LR recombination reaction (Gateway LR Clonase II Enzyme mix, Invitrogen) was then performed using p5E_*cdh5*E, pme_*cdh5*P:Gal4VP16, with pminTol-R4-R2pA to give pminTol- *cdh5*E-*cdh5*P: Gal4VP16. *prmt5* coding sequence from pBluescript II KS+ PRMT5 WT [31] was consecutively cloned into pCS2+ and pme_mcs by using BamHI/EcorI and BamHI/XbaI respectively. Then it was cloned into pDest *insUAS R1R2 CC* using LR clonase II (Thermo Fisher Scientific), to produce *pinsUAS:PRMT5*. 535pb and 430pb of respectively *cdh5* and *fli1a* coding sequences were cloned into PGEM-T vector (Promega) for riboprobes production. Oligonucleotide sequences are listed in S2 Table.

### Generation of *prmt5*⁻/⁻ mutants by CRISPR/cas9

The guide RNA (gRNA) was designed using CHOPCHOP CRISPR Design website [59]. The designed oligos were annealed and ligated into pDR274 after digestion of the plasmid with BsaI (NEB). The gRNA was prepared *in vitro* using the MEGAshortscript T7 transcription kit

(Ambion) after linearizing the plasmid with DraI (NEB) [60] before being purified using illustra MicroSpin G-50 Columns (GE Healthcare). 1 nL of a solution containing 10μM EnGen Cas9 NLS (NEB) and 100 ng/μl of gRNA was injected at the one-cell stage. WT, heterozygous, and homozygous *prmt5* animals were identified by PCR. Oligonucleotide sequences are listed in S2 Table.

## Microinjections

*Tg*(*cdh5*:GAL4VP16);*Tg(UAS:KAEDE)* and *Tg(UAS:PRMT5)* lines was generated using pmin-Tol- *cdh5*E-*cdh5*P:Gal4VP16 and *pinsUAS:PRMT5*, *respectively* by Tol2 transposition as described previously [61]. Control and *prmt5* morpholino oligonucleotides (MOs) were described previously [23]. Embryos from in-crosses of the indicated heterozygous carriers or wild-type adults were injected at one cell stage with 6 ng of MO. pBluescript II KS+ PRMT5 WT and pBluescript II KS+ PRMT5 Mutant [31] were linearized by EcoRI (NEB) and transcribed by T7 (Promega). 200 pg *prmt5WT* mRNA, or *prmt5MUT* mRNA were injected at one cell stage.

## EPZ015666 treatment

Embryos were treated with 5 μM, 10 μM or 20 μM of the Prmt5 inhibitor EPZ015666 (Sigma-Aldrich) or with DMSO from 90% epiboly to 20 hpf (WB) or from 30 hpf to 3 dpf (RT-qPCR).

## RNA extraction, Reverse transcription and real-time PCR

Embryos were dissected at the indicated stage after addition of Tricaine Methanesulfonate. Genomic DNA was extracted from dissected embryo heads to identify their genotype and the corresponding dissected tails were conserved in TRIzol Reagent at -20˚C. After identification of wild- type and mutant embryos, total RNAs from at least 6 identified tails were extracted following manufacturer's instructions (Invitrogen). Total RNAs were converted into cDNA using Prime Script cDNA Synthesis Kit (Takara) with Oligo(dT) and random hexamer primers for 15 min at 37˚C according to manufacturer's instructions. cDNAs were then diluted 20-fold and quantified by qPCR using SsoFast Evagreen Supermix (Bio-Rad) and specific primers. Data were acquired on CFX96 Real-Time PCR detection System (Bio-Rad). Samples were analyzed in triplicates and the expression level was calculated relative to zebrafish housekeeping gene *EF1α*. Oligonucleotide sequences are listed in S2 Table.

## Live imaging

For the transgenic lines *TgBAC(cdh5*:GAL4FF);*Tg(UAS:GFP)* and *Tg(cdh5*:GAL4VP16);*Tg(UAS:KAEDE)*, and *Tg(fli1a:eGFP)* embryos were placed in 1.5% low melt agarose with Tricaine on a glass-bottomed culture dish filled with egg water. Images were acquired using the confocal microscope TCS SP8 (Leica Microsystems) with an L 25 × /0.95 W FLUOSTAR VIZIR objective (zoom X1.25) using the scanner resonant mode. Confocal stacks were acquired every 10 min from 28 to 38 hpf to generate movies.

## Immunostaining and *in situ* hybridization

After fixation or rehydratation, embryos were washed twice with Phosphate Buffered Saline/ 1% Triton X-100 (PBST), permeabilized with PBST/0.5% Trypsin for 30 sec and washed twice again with PBST. After blocking with PBST/10% Fetal Calf Serum (FCS)/1% bovine serum albumin (BSA) (hereafter termed 'blocking solution') for at least 1 h, embryos were incubated with antibodies directed against GFP (Torrey Pine, Biolabs), Prmt5 (Upstate #07405), MEP50/

Wdr77 (Cell Signaling Technology #2823) or Symmetric Di-Methyl Arginine Motif (SDMA, Cell Signaling Technology #13222) in blocking solution overnight at 4°C followed by 5 washing steps with PBST. Embryos were then incubated with the appropriate Alexa Fluor-conjugated secondary antibodies (Molecular Probes) for at least 2 h at room temperature and washed three times. Nuclei were then stained with TO-PRO3 (Molecular Probes) and washed twice with PBST. Embryos were dissected, flat-mounted in glycerol and images were recorded on a confocal microscope as above. *In situ* hybridization was carried out as previously described [35]. Riboprobes were produced by linearizing PGEMT-cdh5 and PGEMT-fli1a vector by SacII and *in vitro* transcription by SP6 polymerase. Embryos were dissected, flat-mounted in glycerol and images were recorded on a confocal microscope as above or using the confocal Zeiss LSM 880.

## Western blots

20 hpf Embryos were lysed in 2x Laemmli sample buffer (4% SDS, 20% glycerol, 125mM Tris-HCl pH 6.5, 0.004% bromophenol blue, 10% β-2-mercapthoethanol). Proteins equivalent to 6 embryos were loaded and samples were separated by SDS-PAGE before being transferred to Nitrocellulose membranes. Membranes were blocked 1h at room temperature (10% milk, 0.1% Tween TBS) and probed at 4°C overnight (2% milk, 0.1% Tween TBS) with α-Symmetric Di-Methyl Arginine Motif (SDMA, Cell Signaling Technology #13222) and α-Tubulin (Sigma-Aldrich #T-6199) primary antibodies. After several washings, membranes were incubated with appropriate horseradish peroxidase-conjugated secondary antibodies for 1h. Detection was performed using Clarity Western ECL substrate (Bio-Rad) according to manufacturer's instruction and signals were analyzed with the ChemiDoc MP Imaging System device (Bio-Rad).

## Chromatin immunoprecipitations

20 hpf dechorionated embryos were fixed in 1.85% formaldehyde in fish water for 15 min at room temperature. Glycine was then added to a final concentration of 125 mM at room temperature for 5 min to quench formaldehyde. Embryos were then rinsed three times in ice cold PBS at 4°C and were snap frozen in liquid nitrogen. Embryos (400 embryos/ChIP) were lysed in buffer containing 10 mM Tris-HCl pH7.4, 10 mM NaCl, 0.5% NP-40 supplemented with protease inhibitors and were then spun at 3500 rpm for 5 min at 4°C. The pellet was resuspended in 100 μl nuclei lysis buffer (50 mM Tris-HCl pH7.4, 10 mM EDTA, 1% SDS + protease inhibitors) and incubated for 10 min at 4°C. Two volumes of IP dilution buffer (16.7 mM Tris-HCl pH7.4, 167 mM NaCl, 1.2 mM EDTA, 1.1% Triton X-100, 0.01%SDS + protease inhibitors) were then added before proceeding to the sonication with a Bioruptor device (Diagenode) to get fragments of 300–500 nucleotides. Samples were spun at 14000 rpm for 10 min at 4°C and the supernatant was transferred to a new tube. Protein A and protein G agarose beads (Sigma-Aldrich) saturated with PBS containing 0.5% BSA were added to the samples for the pre-clearing step for 4 h at 4°C on a rotating wheel. After a short centrifugation, 3 μg of the following antibodies were added to each supernatant: anti-Brg1 (#NB100-2594, Novus Biologicals), anti-PRMT5 (mix of #07–405, Upstate Biotechnology and #MA1-25470, Invitrogen), anti-MED1/PPARBP (#H00005469, Abnova). A mock ChIP with no antibody, was included as a control. After 4 h of incubation with antibodies, protein A and protein G agarose beads were added to the mix and incubation was carried on overnight at 4°C on a rotating wheel. The day after, beads were washed twice in RIPA buffer (10 mM Tris-HCl pH 8.0, 140 mM NaCl, 1 mM EDTA, 1% Triton X-100, 0.1% SDS, 0.1% sodium deoxycholate) then three times in LiCl solution (10 mM Tris-HCl pH8.0, 250 mM LiCl, 1 mM EDTA, 0.5% NP-40, 0.5% sodium

deoxycholate) and finally twice in TE (10 mM Tris-HCl pH8.0, 1 mM EDTA). Proteins on beads were then eluted twice in 10 mM Tris-HCl pH8.0, 1 mM EDTA, 50 mM NaCl, 1% SDS during 15 min at 65˚C. The supernatants were adjusted to a final concentration of 200 mM NaCl before incubation at 65˚C overnight to reverse the crosslink. The day after, RNAse A was added for 1 h at 37˚C before proceeding to proteinase K treatment. DNA was purified by addition of phenol/chloroform/isoamyl alcohol followed by ethanol precipitation. Purified DNA was dissolved in RNAse-free, DNAse-free water (Sigma-Aldrich) and further used for qPCR. Samples were analyzed in duplicates and fold enrichment was calculated relative to the input and to a negative region (gene desert) on chromosome 7 (Active Motif). The final fold enrichment value was then calculated relative to the mock ChIP. Primer sequences are listed in S2 Table.

## Image processing and measurements

Confocal images and stacks were either analyzed with ImageJ software, LAS X or ZEN blue. Nuclei of ISV cells and gata2b+ cells were counted using the Multipoint tool of ImageJ. ISV lengths were measured by drawing a line between the base and the tip of ISV on ImageJ. Contours of the Dorsal Aorta were drawn using the Freehand Selection Tool with a digital pen and the area was then measured. Fluorescence intensity corresponded to the measurement of average gray value for each entire image.

## Statistical analysis

Statistical comparisons of datasets were performed using GraphPad Prism software. For each dataset, we tested the assumption of normality with D'Agostino-Pearson tests and accordingly, unpaired t-test, Mann-Whitney test, One-way ANOVA, two-way ANOVA or Kruskal-Wallis test were used to compare dataset; means and SDs are shown for each graph/plot. The test used as well as the number of independent experiments performed and the minimal number of biological replicates are indicated in each figure legend.

## Bioinformatic analysis

Published single cell data from total embryos at 10 hpf, 14 hpf, 18 hpf and 24 hpf [33] were analyzed using the R package Seurat [62,63]. After data clustering, clusters of endothelial cells from each stage were identified by the expression of several endothelial specific genes (*etv2*, *fli1a*, . . .). Then, we examined the level of expression and the percentage of cells expressing our gene of interest at each developmental stage. ATAC-seq data [38] and ChIP-seq data [6] were inspected using the Galaxy platform [64].

## Supporting information

**S1 Fig. Analysis of hematopoietic lineage markers.** (A) Average number of HSCs enumerated per confocal stack in control and in *prmt5* morphant embryos at 40 hpf. Data are from 2 independent experiments with at least 3 individuals per experiment and a t-test was performed. *** P<0,001. (B, C) Confocal projections of immunostaining with anti-phosphoH3 antibody (in red) of *Tg*(*gata2b*:*Gal4; UAS*:*lifeactGFP*) transgenic embryos either injected with control morpholino (B) or *prmt5* morpholino (C) at 40 hpf. Scale bar 100 μm. (D) Schematic representation of vascular (green) and hematopoietic (red) systems in a zebrafish larva. Bracket indicate the Caudal Hematopoietic Tissue (CHT). (E, F) Confocal projections of CHT from transgenic Tg(*gata2b*:*Gal4; UAS*:*lifeactGFP*) embryos at 3 days in the same conditions as in B-C. Scale bar 100 μm. (G) Average number of HSPCs enumerated per confocal stack in

control and *prmt5* morphant embryos at 3 days from 2 independent experiments with at least 4 individuals per analysis. T-test was performed. *** P<0.001. (H-K) Relative mRNA expressions determined by RT-qPCR in either the head (H, I) or the trunk (J, K) of 3-day old control and *prmt5* morphant embryos, from 4 independent experiments with at least 10 animals per condition. T-test was performed. * P<0,05.
(TIF)

**S2 Fig.** (A-C') Confocal projections of transgenic *Tg(fli1a:GFP)ʸ¹* embryos with endothelial cells (in green) after immunostaining against Prmt5 (in magenta). (A-A") Dorsal view of the lateral plate mesoderm at 14 somite- stage. Yellow rectangle delimits the close up of Prmt5+ endothelial cells (A'-A"). Prmt5+ cells appear in magenta (A-A") and endothelial cells in green (A-A'). Anterior is on top. Scale bars 100 μm (A) and 25 μm (A'). (B, B') Confocal projections focusing on endothelial cells (in green) from the dorsal aorta (DA) and the cardinal vein (CV) at 24 hpf. Red and blue arrows point to Prmt5+ cells (in magenta) from the DA and the CV, respectively. Red and blue lines represent DA and CV diameters, respectively. Scale bar 50 μm. (C, C') Confocal projections focusing on sprouting ISVs (in green) at 24 hpf. Light blue and yellow arrows point to tip and stalk cell, respectively. (D) Schematic representation of the trunk vasculature with ISVs sprouting from the DA. The tip cell leads the cell migration and the stalk cell maintains the connection with the DA. (E) Expression heatmaps for *etv2*, *prmt5*, *MEP50/Wdr77*, *Smarca4a/Brg1*, *MED1* and identified Prmt5 target genes, in endothelial cells at 10 hpf, 14 hpf, 18 hpf and 24 hpf. Data were derived from single cell RNA-sequencing from Wagner et al. [33]. The expression level is colored-coded from absence of expression (green) to highest level of expression (white).
(TIF)

**S3 Fig.** (A, B) Confocal projections of transgenic *Tg(fli1a:GFP)ʸ¹* wild-type (A) and *prmt5* mutant (B) embryos at 48 hpf. Scale bar 100 μm. (C, D) Average number of disconnected ISV per confocal projection (C) and average ISV area in μm2 (D) in control and *prmt5* mutant embryos, from 3 independent experiments with at least 4 animals per condition. Mann-Whitney tests were performed. **** P<0.0001; *** P<0.001. (E-G) Confocal projections of transgenic *TgBAC(cdh5:GAL4FF);*Tg(*UAS:GFP*) and *TgBAC(cdh5:GAL4FF);*Tg(*UAS:GFP*); Tg(*UAS:prmt5*) embryos at 28 hpf. Control morphant is on the top left panel (E), *prmt5* morphant embryos not expressing (F) or mis-expressing *prmt5WT* in blood vessel (G). The fluorescent intensity is colored-coded, from Low intensity (L) in black to High intensity (H) in white. Scale bar 100 μm. (H, I) Average GFP fluorescence intensity per confocal projection (H) and average ISV length in μm (I), for control, *prmt5* morphant embryos not expressing or mis-expressing *prmt5WT* in blood vessels, from 2 independent experiments with at least 3 animals per condition. One-way ANOVA and Kruskal-Wallis test were performed. * P<0.05, *** P<0.001. (J-R) DIC images and confocal projections of wild-type (J,L,O,Q) and *prmt5* mutant embryos (K,M,P,R) after *in situ* hybridization or fluorescent *in situ* hybridization against *cdh5* (J-M) and *fli1a* (O-R). Scale bar 100 μm. (N, S) Percentage of embryos (y axis) presenting a high or a low level of expression of *cdh5* (N) or *fli1a* (S), according to their genotype (x axis), from 3 independent experiments with at least 4 animals per condition.
(TIF)

**S4 Fig. Chromatin profile visualization of endothelial cells from the UCSC Genome Browser.** ATAC-seq peaks as determined by Quillien et al. [38] flanking potential Etv2 target genes as indicated (*cdh5*, *esama*, *fli1b*, *atgr2*, *fli1a*). GFP+ cell track corresponds to endothelial cell ATAQ-seq profile and GFP-minus track represents non-endothelial cell profile. Endothelial cell peak track shows specific peaks for endothelial cells. Promoter regions (P) and putative

enhancers (E) are highlighted in light orange and light pink, respectively.
(TIF)

**S1 Table. RNA-seq in Endothelial cells/non-endothelial cells from Quillien et al. [38] (derived from dataset GSE97256).**
(XLSX)

**S2 Table. Oligonucleotides used in this study.**
(DOCX)

**S1 Data. All numerical data underlying graphs shown in the manuscript.**
(XLSX)

## Acknowledgments

We would like to thank the Blader lab for helpful discussions and support; Dr Blader for graciously allocating space and time to AQ to complete the experiments; Drs. Cau, Male and Navajas Acedo for critical reading of the manuscript; Dr. Renaud (iGReD, Clermont-Ferrand) for help in bio-informatics analyses. A. Laire for excellent zebrafish care; Dr. Tavernier from the CREFRE platform; B. Ronsin, S. Bosch and L. Ligat from the Toulouse Rio Imaging (TRI) platform; M. Aguirrebengoa from the BigA Core Facility, CBI Toulouse; Dr. Herbomel (Institut Pasteur, Paris) for the *Tg(gata2b*:*Gal4;UAS*:*lifeactGFP)* transgenic line.

## Author Contributions

**Conceptualization:** Aurélie Quillien, Laurence Vandel.

**Formal analysis:** Aurélie Quillien, Lucas Waltzer, Laurence Vandel.

**Funding acquisition:** Lucas Waltzer, Laurence Vandel.

**Investigation:** Aurélie Quillien, Guerric Gilbert, Manon Boulet, Séverine Ethuin.

**Methodology:** Guerric Gilbert.

**Project administration:** Laurence Vandel.

**Resources:** Lucas Waltzer.

**Supervision:** Laurence Vandel.

**Validation:** Aurélie Quillien, Manon Boulet, Séverine Ethuin.

**Writing – original draft:** Aurélie Quillien, Laurence Vandel.

**Writing – review & editing:** Aurélie Quillien, Lucas Waltzer, Laurence Vandel.

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
