## [Decision Letter · Decision Letter 0]

5 Jan 2021

Dear Dr Vandel,

Thank you very much for submitting your Research Article entitled 'Prmt5 promotes vascular morphogenesis independently of its methyltransferase activity.' to PLOS Genetics.

We apologize for the delay in considering your paper. The use of Review Commons is new to the Associate Editor on this manuscript and consequently there was some confusion that had to be addressed (unfortunately, the holidays contributed to the delay).

The manuscript has been fully evaluated at the editorial level and by an independent peer reviewer that was commissioned by PLOS Genetics. All reviewers were enthusiastic about this work, but as previously noted with Review Commons, there are substantial concerns. We appreciate that you have begun to address many of the concerns and we are willing to review a much-revised version that takes into account all 4 reviews. We cannot, of course, promise publication at that time.

If you decide to revise the manuscript for further consideration at PLOS Genetics, please aim to resubmit within the next 60 days, unless it will take extra time to address the concerns of the reviewers, in which case we would appreciate an expected resubmission date by email to plosgenetics@plos.org.

[LINK]

We are sorry that we cannot be more positive about your manuscript at this stage. Please do not hesitate to contact us if you have any concerns or questions.

Yours sincerely,

Marisa S Bartolomei

Associate Editor

PLOS Genetics

John Greally

Section Editor: Epigenetics

PLOS Genetics

Reviewer's Responses to Questions

**Comments to the Authors:**

Reviewer #1: Major comments

(1) The expression of prmt5 protein appears to be ubiquitous, and figures 2A-A'', 2B-B', and 2C-C' do little to show that prmt5 protein is expressed in endothelial cells. Even with the provided arrow labelling, I find the intended demonstration of protein expression in ECs hard to appreciate. More importantly, it would be key to show that the gene has a function in endothelial cells - the phenotype shown in 2F, e.g., could stem from delay because of systemic absence of prmt5. Transplanting mutant cells into wild type embryos would demonstrate a specific role of the gene in ECs.

(2) The authors state that prmt5 morphants reproduced the phenotype observed in prmt5 mutants in terms of reduced

ISV length. However, the morpholino leads to a very significant reduction in cell number within ISVs (figure S2C), while muants do not show this (figure 2J). I am refering the authors to the Stainier et al guidelines on the use of morpholinos, and would like to point out that some of the listed controls (such as dose/response curves) have not been carried out. Rather than going into an extensive morpholino QCing exercise, the authors might be better off to replace the MO data by data collected from equivalent experiments in mutants.

(3) Concerning down-regulation of gene transcription in prmt5 mutants, it would be good if the authors were to include a statement about non-EC expression of the genes analysed. Are the genes tested exclusively expressed in ECs? If so, then using whole embryos for analysis is fine. I did a quick check for amotl2a, and, based on ZFIN images, there appears to be significant expression in the brain. Now, it is of course interesting to see amolt2a being regulated by prmt5 as such, but the way the paper is presented implies that the authors focus on the role of prmt5 in the vasculature. However, in figure 2C and when using whole embryos, the authors are really testing overall expression of target genes, also outside the vasculature. The same is true for scl expression in figure 1H, the majority of scl expression has been reported to be outside the hemogenic endothelium.

Minor comment:

The reference list contains some strange looking references, such as the Alestrom, Beacon, Marrass papers. Page numbers are either missing or likely incorrect.

In the introduction, the Isogai 2001 paper certainly does not make the point the authors use it for, a more recent review such as the one from Hogan in Dev Cell or the one from the Affolter lab in 2016 might be more appropriate.

**Have all data underlying the figures and results presented in the manuscript been provided?**

Reviewer #1: None

PLOS authors have the option to publish the peer review history of their article (what does this mean?). If published, this will include your full peer review and any attached files.

Reviewer #1: No

---

## [Decision Letter · Decision Letter 1]

2 Jun 2021

Dear Dr Vandel,

We are pleased to inform you that your manuscript entitled "Prmt5 promotes vascular morphogenesis independently of its methyltransferase activity." has been editorially accepted for publication in PLOS Genetics. Congratulations!

Yours sincerely,

Marisa S Bartolomei

Associate Editor

PLOS Genetics

John Greally

Section Editor: Epigenetics

PLOS Genetics

Comments from the reviewers (if applicable):

Reviewer's Responses to Questions

**Comments to the Authors:**

Reviewer #2: The authors have addressed my concerns adequately. I have no further comments.

Reviewer #3: The authors have satisfactorily addressed my critique.

Reviewer #4: Overall, the manuscript revision is meticulously performed, and several key experiments have been done. As a whole, the authors have successfully demonstrated the importance of Prmt5 in promoting vascular morphogenesis

**Have all data underlying the figures and results presented in the manuscript been provided?**

Reviewer #2: Yes

Reviewer #3: None

Reviewer #4: Yes

PLOS authors have the option to publish the peer review history of their article (what does this mean?). If published, this will include your full peer review and any attached files.

Reviewer #2: No

Reviewer #3: No

Reviewer #4: No

**Data Deposition**

http://datadryad.org/submit?journalID=pgenetics&manu=PGENETICS-D-20-01800R1

**Press Queries**

---

## [Editor Report · Acceptance letter]

17 Jun 2021

PGENETICS-D-20-01800R1 

Prmt5 promotes vascular morphogenesis independently of its methyltransferase activity. 

Dear Dr Vandel, 

We are pleased to inform you that your manuscript entitled "Prmt5 promotes vascular morphogenesis independently of its methyltransferase activity." has been formally accepted for publication in PLOS Genetics! Your manuscript is now with our production department and you will be notified of the publication date in due course.

With kind regards,

Agota Szep

PLOS Genetics

On behalf of:
